# Multifunctionality of Forests: A White Paper on Challenges and Opportunities in China and Germany

**J. Philipp Benz** [1,*], **Shaolin Chen** [2,3], **Shuangren Dang** [4], **Matthias Dieter** [5], **Eric R. Labelle** [1], **Guangzhe Liu** [6], **Lin Hou** [6], **Reinhard M. Mosandl** [1], **Hans Pretzsch** [1], **Klaus Pukall** [1], **Klaus Richter** [1], **Ralph Ridder** [7], **Shuaichao Sun** [6], **Xiaozhou Song** [6], **Yifei Wang** [6], **Hongli Xian** [8], **Li Yan** [6], **Jie Yuan** [6], **Shuoxin Zhang** [6] and **Anton Fischer** [1]

[1] TUM School of Life Sciences Weihenstephan, Technical University of Munich, 85354 Freising, Germany; eric.labelle@tum.de (E.R.L.); mosandl@forst.wzw.tum.de (R.M.M.); Hans.Pretzsch@tum.de (H.P.); klaus.pukall@tum.de (K.P.); richter@hfm.tum.de (K.R.); antonf@t-online.de (A.F.)

[2] College of Life Science, Northwest A&F University, Yangling 712100, China; slc1916@icloud.com

[3] Biomass Energy Center for Arid and Semi-arid Lands, Northwest A&F University, Yangling 712100, China

[4] Forestry Academy of Shaanxi, Xi'an 710082, China; dang_shr@sina.cn

[5] Thünen Institute of International Forestry and Forest Economics, 21031 Hamburg, Germany; matthias.dieter@thuenen.de

[6] College of Forestry, Northwest A&F University, Yangling 712100, China; gzl66106@nwsuaf.edu.cn (G.L.); houlin_1969@nwsuaf.edu.cn (L.H.); sun_sc@yeah.net (S.S.); songxiaozhou@nwsuaf.edu.cn (X.S.); wangyifei1223@zafu.edu.cn (Y.W.); liliyan@nwsuaf.edu.cn (L.Y.); yuanjie@nwafu.edu.cn (J.Y.); sxzhang@nwsuaf.edu.cn (S.Z.)

[7] FEDRC GIZ Forest Policy Facility, Beijing 100125, China; Ralph.Ridder@giz.de

[8] Center for Natural Forest Protection Program in Shaanxi Province, Xi'an 710082, China; xianhla@163.com

**\*** Correspondence: benz@hfm.tum.de

**Abstract:** Both in Germany and in China, there is strong expertise regarding the different aspects of forest management, as well as forest products management. Nevertheless, forestry in both countries is facing challenges, some of which are regional, but many of which are shared. Therefore, experts from both countries (Technical University of Munich Germany; Northwest A&F University Yangling, China; Forestry Academy of Shaanxi, China; Thünen Institut, Germany; FEDRC GIZ Forest Policy Facility (Forestry Economics Development and Research Center of the Deutsche Gesellschaft für Internationale Zusammenarbeit GmbH), Germany; and Center for Natural Forest Protection in Shaanxi, China) met to share their knowledge and deduce recommendations for future multifunctional forest management for the temperate zone. The workshop, held at the Northwest A&F University in September 2018, included presentations and intensive discussions, as well as a field tour. The results of the workshop that are summarized in this white paper are meant to provide an overview of the multi-faceted nature of the topic for interested scientists and forest practitioners, describe tools that can be used to analyze various aspects of multifunctionality and, in an exemplary fashion, highlight gathered experience from long- and short-term experiments. Included are social demands, economic goals, and scientific baselines. The topics reach from economic evaluations of forest ecosystem services over forest management practices, including afforestation, restoration, and preparations to face climate change, to wood/forest products utilization and participation of local people for poverty reduction. Overall, an optimistic picture emerges, showing that by using adapted forest management practices, which try to embrace the concept of multifunctionality, various use schemes and demands can be integrated at single sites, allowing us to achieve both environmental protection and productive forests, including societal demands, as well as aspects of tradition and national identity.

**Keywords:** multifunctionality; ecosystem services; forest management; forest products; societal demands

---

## 1. Background

After the end of the last ice age, only ~8000 years ago, around 62 million km$^2$ (41.6%) of the world's mainland was covered by forests. Since then, human pressure on forests has been continuously increasing. By now, nearly half of the world's native forests have been converted to farmland, pastureland, or other uses, and a large proportion of the remaining forests are in intensive use. In fact, only one fifth of the original forests still exist as large, relatively natural ecosystems [1].

Germany, as an example, would be almost entirely covered by forests without human intervention under recent climatic conditions. Nevertheless, forest cover had been reduced to less than 20% in medieval times [2]. Following intense re-development and restructuring, although strongly reduced in tree species richness, nowadays, forests cover approximately 32% of the German landscape [3], but are subjected to an intense utilization pressure. In China, the process of de- and re-forestation was similar, but the reduction in tree species diversity and deforestation began earlier, and reforestation efforts only started to increase in the late 20th century.

What are the reasons that forests were and still are under such pressure? Firstly, forested land in many cases has been converted to arable fields or pastureland for food production. The conversion of tropical rain forests in Southeast Asia into palm oil and/or rubber plantations is an extreme example but is not covered here. Secondly, forested areas were and still are used for settlement and infrastructure, a process of increasing importance today. Thirdly, the forests themselves offer important resources, such as timber, which for millennia has been the main energy carrier (firewood for households and developing industries) and has been used as quality wood in the form of dimensional timber (lumber) for construction, furniture, fences etc. While the first two aspects led to a long-term loss of forest area, the third led to a strong change in the structure of the remaining forests through several management practices, such as coppicing, or by preference of special species, such as those introduced from other ecosystems (e.g., *Picea abies*, Norway spruce, in large parts of central Europe) or from other continents (e.g., *Pseudotsuga menziesii*, Douglas-fir, in central Europe, and *Robinia pseudoacacia*, Black Locust, in parts of central China).

In the last few decades, it became increasingly obvious that timber is not the only resource offered by a forest ecosystem. There are many other resources and ecosystem services offered by forests, such as:

(1)　Balancing the water regime of a region by storing water in rainy periods and releasing it in periods of drought, thus buffering both flood and drought events. The Chinese National Forest Protection Program (NFPP), being one out of six Chinese "key ecological construction programs", is one such protective measure.

(2)　Prevention of soil erosion, which is especially important in landscapes prone to erosion, such as the Loess Plateau of Central China. The Sloping Land Conversion Program (SLCP) is another of the six Chinese "key ecological construction programs" that was designed to counteract this risk factor.

(3)　Protection of wildlife habitats and natural biodiversity, which, on the one hand, has to be seen as a natural heritage and, on the other hand, may become useful in the future, e.g., as a genetic resource. The Rio process for biodiversity protection is an important outcome, formalised, e.g., in Europe in the EU Fauna-Flora-Habitat directive and its instrument, called "NATURA 2000".

(4)　Contributing to fight against climate change by absorbing $CO_2$ and storing its carbon in the form of timber for decades or even centuries. This $CO_2$-mitigation is now a task of global relevance.

(5)　Acting as land for recreation. This is an increasing demand in many forest areas, particularly those close to megacities.

All of these demands can either be realized by separating existing forest areas and declaring parts of them as target areas for one (or a few) of these goals and leaving other areas unmanaged. Alternatively, another approach is to try to find ways to fulfill as many demands as possible at each single landscape patch, thereby realizing true multifunctionality of forests. On the one hand, forest

management methods and technologies have to be developed and optimized to reach all these goals; a task for forest scientists and forest practitioners. On the other hand, society has to rank the demands according to their degree of priority; this is why stakeholders of such communities (policy makers) and experts dealing with decision processes in forest policy also have to be involved in an active discussion.

Since forests naturally offer a wide range of goods and services that are helpful for humans, the challenges often arise when trying to take advantage of as many of these as possible, while at the same time preserving the system, so that these goods and services can continue to be offered for a long time. To find a balanced consideration between the many demands is complex and is becoming ever more complicated with the rapid rate of climate change. The results of this are already influencing ecosystems (forests and beyond) on a global scale. Because forestry is dealing with trees, which are organisms with a lifespan of decades or centuries, it is essential for forest planners to have far-sighted knowledge on what could happen within the rotation of each forest stand. It is therefore necessary to include researchers in the discussion who deal with prediction scenarios of future living conditions for trees, as well as scientists who are able to translate the abstract knowledge on such scenarios into consequences for tree and forest stand growth under such changing conditions (modelling of forest structure for growing stands under different climate conditions). In addition, the economic value(s) of all possible forest functions need to be assessed and communicated correctly.

Finally, when planting trees to create a new forest stand, it is vital to know how and for which purpose the timber (or other products, including non-timber forest products) should be used. Therefore, the selection of adequate species and the implementation of appropriate forest management practices and associated operations are essential from the very beginning of the planning process.

The topic of forest multifunctionality is of highest importance and relevance for more or less all countries on planet earth. However, in order to achieve it, a number of challenges need to be addressed, some of which are general, and some have to be locally adapted. This white paper reports on the results of a dedicated Sino-German workshop "Multifunctionality of Forests—A Key to Adaptive Forest Management", including about 20 speakers and a total of 40 attendees, highlighting the importance of multifunctionality in forestry and the suggestions made to reach this goal under consideration of the current local conditions.

**Terms of Importance**

**Multifunctionality/ecosystem services**: The term "forest function" was introduced and promoted by Victor Dieterich [4], a forest scientist who worked at the forestry school of the University of Munich for a long period. The term highly influenced forest policy development in Germany [5] despite not having a clear definition. On the one hand, forest functions are defined as interests of society and, on the other hand, as positive impacts of natural processes within forests on society. The distinction between goals of society or special societal actors, natural processes, ecosystem functions, and services is better synthesized in the "ecosystem services" concept [6]. Nevertheless, there is an ongoing debate about a clear distinction between the mentioned terms [7]. However, the concept of ecosystem services must not be restricted to the ecosystem itself; it requires both an understanding of ecosystem processes as well as of societal dynamics and valuations. Especially, the cultural services cannot be understood only by the help of analyzing ecosystem elements or processes, but by understanding the meaning which is attributed to objects by society. These objects can be called ecosystems, nature, landscape, wilderness etc. [8,9].

Because forested land is limited on earth, it is better to address more than one function to each forest site simultaneously, as long as the functions are not contradicting. As such, "multifunctional land use" is realized. Multifunctionality therefore denotes the phenomenon that the landscape actually or potentially provides multiple material and immaterial "goods" that satisfy social needs or meet societal demands by the states, structures, or processes of the landscape. As Wiggering et al. [10] defined, multifunctionality addresses "environmental and economic services as long as society expresses a demand for them". This also clearly highlights that multifunctionality is dependent on current and local interests and can therefore differ greatly around the globe.

The definition of multifunctionality by the World Trade Organization (WTO) Glossary (for agriculture, [11]) is: "the idea that agriculture has many functions in addition to producing food and fibre, e.g., environmental protection, landscape preservation, rural employment, food security etc." In this way, multifunctionality could be called "non-trade concerns" of agriculture. However, in the context of forestry, the broader definition probably fits better, since the production of wood is one of the major "functions" or "tasks" of forests.

Thus, the most important "tasks" or "functions" of forests for society are to produce and/or offer high quality timber, firewood, non-timber forest products, fiber, groundwater, potable water, clean air, and buffering capacities for matter and energy, such as noise, landscape structuring (e.g., counteracting erosion), water (floods vs. drought), carbon (sequestration to counteract climate change), offering habitats for both humans (recreation) as well as wildlife (protecting biodiversity) and offering jobs and income.

**Sustainability:** The idea of sustainability dates back to medieval times. Sustainable practices, such as active seed production or coppices systems, were developed in Europe (e.g., Peter Stromer in Nürnberg, Germany; [12]). The term "sustainable" was first coined in German (i.e., "nachhaltig") in a textbook on silviculture, slightly more than 300 years ago, by Hannß Carl von Carlowitz [13]. He was the head of the silver mining industry in the former Kingdom of Saxony (now a Federal State of Germany) and realized that more wood was being used in the silver mining industry at that time than was regrown. Due to the leading role of German foresters in developing modern forestry, the term spread in the community. Nowadays, the idea is much wider, covering all aspects of human use of nature, condensed in the definition in the UN Brundtland Report [14]: "Sustainable development is development that meets the needs of the present without compromising the ability of future generations to meet their own needs". However, it has to be noted that sustainable utilization of natural resources does not necessarily mean to mimic natural processes. In contrast, nature very often works unsustainably, leading to population collapse or species extinction. Sustainable use, therefore, is a principle introduced by humans to prevent such a collapse of the human population.

## 2. Analyses and Case Studies

In the following, short commentaries and case studies are presented that address the main discussion points regarding multifunctionality of forests.

### 2.1. Societal Demands: Multiple Actors and Multiple Demands on a Limited Resource—A Friction Zone

2.1.1. Analysis of Different Forms of Integration of Multiple Interests in Forestry

In the political debate about multifunctional management, the terms of integration and segregation are of high importance, especially when there is a debate about conflicting interests, such as wood utilization vs. protection of biodiversity (e.g., dead wood). Forestry actors often favor the concept of priority areas to address these conflicts [15].

"Inclusion" means that different interests can be fulfilled on the same spot without conflicts. For centuries, coppice with standard systems were an example for the inclusion of interests for firewood, timber, tannery, litter, and pig fattening. Integration occurs if different interests cannot be fulfilled optimally at the same spot, so that spatial or temporal prioritization of one interest over the other(s) is needed. There are no clear boundaries between areas of prioritization. Starting with the integration level, there is a clear power imbalance between actors with different interests. The powerful actors decide if they integrate, segregate, or exclude other interests within "their" forests. In European alpine forest pasture systems, grazing is integrated into forestry. There is a fluent transition from dense forest stand to light sheltered woods to open pastures. A segregation clearly separates the interests, which cannot or should not be fulfilled on the same area. Reorganization of forest pasture systems often relies on this concept. Livestock is concentrated in (fenced) areas with a higher degree of open pastures and less steep terrain, while protection forests are released from the grazing pressure. Forests within

fencing are less profitable for the forest owner, and the interests of the farmers are dominant. Exclusion means that certain functions are completely suppressed by the dominant function within the area of interest. The Bavarian Forest Rights Law (BFRL) mentions the possibility of grazing rights being transferred to non-forested areas. The forest owner would have to provide such pastures outside the forests. The last possibility is that certain interests are eradicated (extinction). The BFRL, for example, allows to directly transform grazing rights into timber (or other) rights. The forest owner also has the possibility to purchase the grazing right. Thus, multifunctional management is a flexible answer to the question, "Who is integrating which interests to which extent on which spatial level into what?" This question is answered within the societal or political discourse, as well as by management decisions of forest owners.

Within the discourse on climate change mitigation and adaptation, new demands, such as carbon storage or mitigation of temperature extremes, are formulated. Even the old coppice systems, which were mostly abandoned in Middle Europe during the 19th century, are reinvented as short-rotation plantations on agricultural land as a means of climate change mitigation. The multifunctional evaluation of the different options is rather complex [16] and is dependent on system boundaries. Deckmyn et al. [17] compared the benefits for carbon sequestration of afforestation with an oak–beech high forest with a poplar short-rotation coppice. While the primary production of an oak–beech forest is much lower (2.5 t C ha$^{-1}$ yr$^{-1}$ after 150 years vs. 6.2 t C ha$^{-1}$ yr$^{-1}$), the total C pool (living biomass, wood products, and soil) after 150 years doubles the C pool of the coppice (324 t C ha$^{-1}$ vs. 162 t C ha$^{-1}$). Regarding energy substitution, coppice reduces emissions four times more than the high forest (24.3–29.3 t $CO_2$ ha$^{-1}$ yr$^{-1}$ vs. 6.2–7.1 t $CO_2$ ha$^{-1}$ yr$^{-1}$).

With the concept of agroforestry, the clear segregation between forests and agricultural land, which is an outcome of the 18th/19th century development of forestry and agricultural sciences in combination with the liberal discourse [5], might change to fulfill the increasing demands formulated for the land use systems within the climate change discourse.

### 2.1.2. Economic Evaluation of Forest Ecosystem Services

In the concept of ecosystem services, according to the Millennium Ecosystem Assessment (MEA, [18]) functions are grouped into the four categories: (i) supporting, (ii) provisioning, (iii) regulating, and (iv) cultural services. These services are crucial constituents of human well-being. Biodiversity is not a service in itself. It forms the basis for all kinds of ecosystem services and therewith for human well-being.

Due to the complexity of forest ecosystems, their services do not necessarily act in the same direction. Albeit some services can support themselves, allowing for a simultaneous improvement of all, others can also be mutually neutral or in some cases conflicting, which means that one ecosystem service can be improved only at the expense of others.

In this field of different interactions, economic valuation can be an appropriate means to find optimum forest management options, in particular from a national perspective. Yet, economic valuation requires profound understanding of the trade-offs between different ecosystem services, as well as comprehensive competence in valuation methods. As only a smaller part of forest ecosystem services are allocated by markets, in particular, valuation approaches for so called "public goods" become a relevant part of economic valuations.

An example from Germany may illustrate the main elements of such an economic valuation and the potential outcome. The study [19] refers to the entire forest area in Germany and focusses on the following three ecosystem services: (i) timber production (provisioning), (ii) carbon sequestration in both standing forest stock and harvested wood products (regulating), and (iii) recreation (cultural). Four different forest management scenarios are analyzed and compared to a baseline scenario: 0) baseline: current economic and legislative framework; 1) postponing harvest age; 2) decreasing harvest age; 3) augmenting set-aside areas; 4) overall reduction of harvests.

Forest development and timber supply, as well as carbon stock changes in forests and harvested wood products, are calculated by means of two specific models: WEHAM (from German: "Waldentwicklungs- und Holzaufkommensmodellierung" and WoodCarbonMonitor [20], respectively. The common unit for the economic valuation is changes in income [21]. For timber production, income changes are gauged on the basis of a physical input-output-table [19], depicting the entire German wood fiber flows, combined with sector-specific added value data. The value of carbon sequestration is calculated on the basis of two alternative approaches: "market" prices and avoidance costs. The cultural service is estimated on the basis of a representative internet-based choice experiment conducted among the German population [22].

Despite several crucial model assumptions and constraints, the results show that the applied new approach allows for a joint valuation of private and public goods. Therewith, a monetary value can be assigned to multifunctional forests, providing a bundle of private and public goods. The applied economic valuation shows to what extent the joint value of the analyzed forest ecosystem services exceeds the value of mere individual services. This is a strong argument for multifunctional forestry in political debates. Moreover, on a forest enterprise level, it helps to find the optimum forest management under different options for multifunctional forestry. Interestingly, for Germany, the baseline scenario proved to be the most favorable from a holistic angle.

### 2.1.3. Introducing Multi-Functionality to China's Forests

Between 1990 and 2015, China increased its forest area by 55 million ha [23]. Today, China's forests cover an area larger than 210 million ha, and 120 million ha are foreseen to be afforested in the next three decades [24]. Existing forests are degraded, with an average standing stock of about 91 m$^3$/ha (in comparison to 340 m$^3$/ha in Germany) [25] and an average annual harvesting volume of about 0.4 m$^3$/ha (in comparison to 8 m$^3$/ha in Germany) [26]. In the past decade, China's forests went through a major tenure reform from the previous state management of all forests to now, with about two thirds of the forests being managed by communities or individuals [25].

Until now, China's forests are still predominantly mono-functional, i.e., there is no established multi-functional forest mapping. About one third of the forest areas are "commercial forests", typically fast-growing tree plantations and mainly same-age mono-species plantations under clear-cut management. About two thirds of the forest are so-called "natural forests", in which logging is prohibited due to a logging ban. Natural forests are increasingly managed, applying the close-to-nature concept in order to stabilize stands and make them less vulnerable to wind, ice storms, calamities, and fire by conversion to mixed-species and mixed-age stands. The objective is ecological forest rehabilitation and a major increase in standing stock, resulting in an expected above-soil carbon sequestration of about 15 billion tons.

Natural forests mainly have three functions: There are "public benefit forests", which can be used for non-timber purposes by the local population, such as for forest grazing, mushroom production, etc; there are "protection forests", e.g., for water catchment protection or for avoiding soil erosion; and there are "tourism forests" for recreation.

In China, with about 1.4 billion people and a high demand for wood and fiber, mono-functionality limits the forest use options. A situation where multiple actors with multiple demands compete for a limited mono-functional forest area suggests that multi-functionality is a more adapted concept for China. Therefore, the Chinese government is in the process of revising its national and province forest policy frameworks towards multifunctional and more sustainable forest management. This process is supported by the BMEL (German Federal Ministry of Food and Agriculture) co-funded Forest Policy Facility, implemented by FEDRC (Forestry Economics Development and Research Center) and GIZ (Deutsche Gesellschaft für Internationale Zusammenarbeit GmbH), offering German forestry expertise to China's National Forestry and Grassland Administration, enabling a Sino-German forest policy dialogue and mutual exchange of experience.

In summary, China's forest managers are facing these challenges: Digesting the forest tenure reform, introducing close-to-nature sustainable management, introducing multi-functionality, and continuing with afforestation of an average of 5 million ha per year. This is driven by a considerable change of policies and, consequently, by a change of forest management practices. It will result in a change of forest management "culture". To ease this process, the Chinese government strongly subsidizes its forests, e.g., in 2017, with 28 billion EUR available for domestic forestry. This change process is expected to carefully prepare China for eventually lifting its logging ban in natural forests and for slowly but steadily increasing its sustainable logging volume on the way to possibly becoming a net wood exporting country.

## 2.2. Forests as Limited Resource Demand Balanced Use Schemes and Sustainable Management Practices

### 2.2.1. Eco-Efficient and Financially Feasible Forest Operations

The level of mechanization in modern forest operations is on the rise in many countries. They commonly rely on machines that weigh between 15 to 40 metric tons to harvest and extract the wood from the forest stand to the truck and accessible roads [27]. To facilitate their movements within a stand, machines are operated on so-called machine operating trails, referring to cleared corridors in the forest that are normally spaced by 20 to 40 m. In the Federal State of Bavaria, Germany, approximately half of the wood volume harvested is with mechanized systems, and this proportion has been increasing slowly due to higher system productivity, ergonomics, and work safety, as compared to motor-manual operations [28]. Because of their high gross mass, traffic frequency, and particular operating environment (un-bound surfaces), forest machines can cause severe soil disturbances, often presented in the form of soil compaction [29] and/or soil displacement [5,30,31].

Detrimental impacts to the physical environment can require several years to decades to naturally rehabilitate [32,33]. For this reason, developing techniques aimed at soil and stand protection are crucial for the health of managed forest ecosystems. Dedicated traffic on pre-defined machine operating trails and operating machines on dryer soil conditions are some prevailing examples. Ultimately, astute pre-planning and the appropriate selection of harvesting systems remain the most value-added tools for successful and less harmful operations. Use of harvesting residue (tree limbs, tops, and foliage) in the form of brush mats that are placed on the machine operating trails are a great example of best management practices aimed at reducing the stress on un-bound forest soils. Despite expanding bioenergy markets and their increased demand for forest biomass, leaving 15 to 20 kg/m$^2$ of brush on trails has been proven to lower disturbances through a reduction of peak pressures exerted by forest machines [34].

Within a scope of multifunctionality, the performance of eco-efficient harvesting hinges on a comprehensive knowledge of soil conditions at the planning stage, machine and soil interactions, wood flow dynamics, and soil and plant relationships. Whenever possible, research projects or field initiatives should consider a more holistic approach to obtain a broader understanding of operational issues in the context of a changing climate and forest [35]. For example, with the increased share of deciduous or mixed species stands, complexity of forest operations is on the rise and new harvesting systems that are tailored at larger diameter stems with more complex morphology are required, since those conditions are vastly different than the homogeneous coniferous species that the systems were originally designed for. A successful example of this type of operation was demonstrated by Labelle et al. [36], where single-grip harvesters operating in >35 cm diameter at breast height beech stands were able to achieve an average harvesting productivity in the range of 29 to 43 m$^3$/PMH$_0$ (Productive Machine Hour excluding delays). Another avenue of growing interest with high potential for value added is the improvement of supply chain management through the continuous use of harvester data. If harvester data (production, positioning, fuel consumption) collected from on-board computers is made available and integrated into existing roundwood supply chains, stakeholders should expect a benefit from increased flexibility, improved logistics, and reduced operational costs.

2.2.2. Effects of Selective Thinning and Residue Removal on Ground Layer Structure and Diversity—A Case Study in a Mixed Pine–Oak Stand of the Qinling Mountains

The role of disturbances on plant species richness has been a central theme in ecology [37,38]. Forest thinning is one of the most efficient tending measures and it is also an effective management technique [39,40]. This management is a major source of disturbance in cultivated, commercial forests [41]. A common thinning practice in China typically includes retention of all post-treatment residues on the ground to maintain soil fertility [42], yet this practice may bury the understory vegetation [40] and potentially decrease herbaceous layer diversity either directly or through increased risk of forest fire. Plant species in the forest ground layer are the main food resources for animals and other organisms in the Qinling Mountains, China. Optimizing a range of understory measures (i.e., species richness, evenness, and cover) is therefore desirable.

Although a considerable number of studies have reported effects of forest thinning on understory vegetation, most have focused on a single factor, such as thinning intensity [41,43–47], composition of tree species [48], fire [43], or fertilization [47]. However, the concurrent effects of two or more factors simultaneously, e.g., the effect of thinning and residual removal on the ground layer, have rarely been explored.

To optimize forest thinning programs, different selective thinning and residue removal regimes were examined over two years in a mixed pine–oak stand to analyze how these influence ground layer cover and diversity [49]. The treatments resulted in an increase in species number, genus, and family, compared with the starting point. Highest richness index, evenness index, and ground cover layer were achieved at a selective thinning intensity of 17–25% and a concurrent residue removal rate of 43–69% [49]. Overall, our results suggest that a medium thinning intensity and thinning residue removal rate may benefit ground layer richness, whereas a high thinning intensity and medium thinning residue removal rate may maximize evenness and ground layer cover.

To conclude, it appears to be very possible to combine the two forest ecosystem functions of wood production and species diversity conservation by operating an optimized thinning program.

2.2.3. Functions of Coarse Woody Debris—A Case Study in the Qinling Mountains

Coarse woody debris (CWD) can be produced under conditions of growth competition between trees, natural death of forests at old ages, natural interferences (e.g., wind, rain, snow, fire, lightning, insects, debris flow, and invasion of fungi), and human interferences (logging, hacking trees) [50]. As an ecological unit, CWD plays an essential role in productivity, nutrient cycling, carbon sequestration, community regeneration, and biodiversity [51–53]. The precise understanding of CWD characteristics is the basis of CWD research and is beneficial for revealing the relationship between the structure and function of CWD. The long-term dynamics of CWD characteristics not only affect the release rates of chemical elements from the CWD, but also influence the diversity of attached vegetation, insects, and microorganisms in CWD [54].

The Qinling Mountains in central China provide an important climate boundary between the southern subtropics and the north temperate zone. The region is distinguished by its high plant and animal diversity. However, in recent decades, the Huoditang Forest Region in the Qinling Mountains has been constantly affected by extreme weather (strong winds), insects, and diseases (*Dendroctonus armandi*), resulting in increased CWD quantity of the main species, *Pinus armandi* and *Quercus aliena* var. *acuteserrata*.

Permanent plots were established (*P. armandi*, *Q. aliena* var. *acuteserrata* forests and mixed forests of the two species) to study CWD dynamics from a long-term project (1996–2016) at the Qinling National Forest Ecosystem Research Station (QNFERS). Forest biomass and CWD characteristics, such as decay states and diameter classes, were quantified during this period and correlated with stand, site, and climatic variables [55]. The forest biomass was dominated by live tree biomass (88%), followed by CWD mass (6–10%). Significant differences in average annual increment of CWD mass were found

among forest stands of different species. Over time, the annual mass of different CWD characteristics increased linearly from 1996–2016 across all forest types.

Our study revealed that CWD varies between forest types and site types, and it also varies over time, depending on disturbances. The high value of CWD for nutrient cycling, water storage capacity, and biodiversity shows that CWD should not only be seen as "lost timber" but as a strong issue to improve forest functioning. Forest management should include CWD in the planning and operating process to optimize many forest functions, for example, nutrient and water storage, erosion prevention, and, last but not least, high quality timber production.

### 2.3. Forests as a Slow-Growing Resource Demand Far-Sighted Knowledge Build-Up

#### 2.3.1. Evaluation of Competition Indices in Modelling Individual Trees

Forest growth and yield models provide foresters the ability to predict future conditions of the forest. Individual tree models have the highest level of resolution and inherent flexibility to project tree and stand development over time for combinations of species mixtures, stand structures, and management regimes [56,57]. This makes individual tree models the ideal tool for management of multifunctional forests. Meanwhile, effects of competition play a key role in modelling individual trees, as they reflect the resource supply of individual trees [58]. Competition indices, mathematical formulations that represent how much an individual tree is affected by its neighbors, can be classified as distance-independent and distance-dependent based on whether tree coordinate locations are required [59]. In many cases, distance-independent indices are preferable for practice, since they are less demanding in data collection and easy to calculate. Moreover, multiple studies suggested that distance-dependent competition indices showed little advantage over distance-independent ones [60–63]. Although a number of studies have involved comparisons of various competition indices (e.g., [62,64]), no clear trends have emerged from these analyses. In addition, most studies applied the same competition index for each component of an individual-tree model (e.g., [65–67]). However, response to competition from each prediction goal may be different. This led to the following question: Does the same competition index perform equally well for different components? Or should different competition indices be used for different prediction goals?

The question was approached by evaluating the inclusion of six distance-independent competition indices in models that predict tree survival and diameter growth [56]. The competition indices were classified into three families: (1) size ratios, which include diameter ratio and basal area ratio; (2) relative position indices, which include basal area of larger trees (BAL) and tree relative position, based on the cumulative distribution function (CDF); and (3) partitioned density indices, which divide the stand density index and relative density into components for individual trees. Results indicated that different families of competition indices were suitable for different tree survival or diameter growth prediction tasks. The diameter ratio was superior for predicting tree survival, whereas both of the relative position indices (BAL and CDF) performed well for predicting diameter growth, with CDF being the best.

For the purpose of multifunctional management, uneven-aged and mixed forests would be smart choices. In this context, individual-tree models with due consideration of competition indices occupy an irreplaceable position. The model set with a suitable competition index for each component will give more accurate prediction for tree survival and diameter growth, which could be the cornerstone for effective management of multifunctional forests. In addition, exploring effects of competition indices may provide useful information for the design of afforestation.

#### 2.3.2. Long-Term Experiments as a Quantitative Basis for Management of Multifunctional Forests

Long-term experiments are an essential quantitative basis for management of multifunctional forests [68,69]. The founding fathers of forest science established the first long-term experiments in the middle of the 19th century [70] and some of them are continuously surveyed to the present day. They

are irreplaceable for long-term bio-monitoring, development, and training of silvicultural guidelines, growth model parameterization, and multifunctional forest planning [71].

Advanced national and regional forest inventories are often seen as a substitute for long-term growth and yield experiments as they may provide representative data on large spatial scales and, in case of repeated measurements, data over longer time periods. Their unique information potential can be exploited with sophisticated big data methods (e.g., [72]). In this way, forest inventories can provide valuable information about the status quo and development of forests on a statistical basis. Thus, inventory data is just right for initializing forest growth models for large-scale simulations and scenario analyses. However, long-term experiments by far outperform the information potential of inventory plots in terms of forest structure and function. They provide insight into mechanisms and cause-effect relationships of forest stand dynamics, which cannot be derived from inventories. Inventories just reflect how forests develop, without any controlled treatment variation. In the case of very common temporary inventory plots, it is impossible to obtain real-time series of forest stand dynamics independence of various treatment variants. Long-term experimental plots, in contrast, mostly comprise un-thinned and otherwise untreated plots as an unambiguous reference and the stand dynamics on such reference plots can be used for quantification of the growth reactions on the treatment plots. By measuring the remaining stand, as well as the removed stand, in detail in defined measurement intervals of 5–10 years, long-term experiments provide the actual total production at a given site, which is most relevant for examining the relationship between site conditions and stand productivity. Experimental plots that comprise various silvicultural treatments may reveal the relation between stand density and productivity, which is most essential for sustainable stand management. Thus, long-term experiments may reflect the effects of thinning and species mixing on stand structure, production, and carbon sequestration. Furthermore, long-term experiments document the species' long-term behavior and any growth trends that may be caused by human influences (e.g., acid rain, over-exploitation, climate change, nitrogen deposition) on forest dynamics [73,74]. Due to their multiple potential of fact-finding, long-term experiments should not be sacrificed thoughtlessly for short-sighted cost-cutting measures. Given the global environmental change and the resulting challenges for sustainable management, the existing networks of long-term experiments should rather be extended in terms of experimental factors, recorded variables, and inter- and transdisciplinary use for science and practice.

Newly established long-term experiments should cover the most important species mixtures in a systematic way. Besides un-thinned monospecific and mixed plots, such experiments should include variants of density reduction, mixing proportions, and mixing patterns. In addition, long-term experiments should cover non-native tree species, agroforestry systems, new silvicultural approaches, such as transformation from homogeneous monocultures to complex uneven-aged mixed stands, and natural stand regeneration approaches. Along with the gradual extension of the sustainability paradigm to a broad range of ecosystem services [75], the variables to be recorded on experimental plots should be extended to wood quality, non-wood forest products, carbon sequestration, biodiversity, habitat properties, and recreation value, and in a psychological context, even spiritual characteristics. Extending the measurement and assessment of existing plots to such characteristics increases their value for monitoring silvicultural practices.

2.3.3. Sustainable Utilization of our Natural Resources in the Face of Multifunctionality in Agriculture and Forestry

It is expected that by 2050, global consumption of food and energy will double, while climate change will have an impact on both crop yields and arable land area [76]. Efforts to increase production of bioenergy helps to increase terrestrial carbon stocks, decrease fossil fuel consumption, and reduce greenhouse gas (GHG) emissions. Northwest China is a typical arid and semi-arid region, like other regions along the Silk Road, with abundant and unique natural resources, including perennial grasses

and forests, as well as vast marginal lands, including sand land, alkaline land, and bare land, suitable for large-scale development of biomass feedstocks [77,78].

Sustainable development of bioenergy and bio-based economies in Northwest China and other arid areas calls for systems approaches that integrate different dimensions. To meet this challenge, Northwest A&F University established the Biomass Energy Center for Arid Lands (BECAL) in 2014, which serves as a platform for systems approaches for biomass feedstock development and conversion technologies. Our systems approach for the economic and ecological dimension of bioenergy and bioeconomy development includes land allocation among competing uses, coupled with the adoption of conservation technologies, incentivized by cost-effective policies [76]. The systems approach requires understanding: (1) environmental impacts and economic costs of cropping allocation and alternative land use practices, (2) synergies and trade-offs between ecosystem and social systems, and (3) market signals, policy, and behavior factors that affect producers to embrace conservation technologies [76,79,80].

In addition to the economic and ecological dimensions of sustainable development, our systems approaches include crop and forest tree improvement for stress-tolerance and environmental benefits [81,82], as well as development of green and cost-effective biorefinery technologies [83]. BECAL has thus been focusing on: (1) ecological and economic assessment of feedstock development, particularly allocation and use of marginal lands [84]; (2) agronomy, stress physiology, and a molecular breeding-based systems approach for the development of next generation biomass feedstocks, e.g., switchgrass and poplar [78,85,86]; and (3) green biorefinery for sustainable and highly-integrated processing of green biomass into multiple products, which includes biological and thermochemical conversion of biomass to biofuels and bio-based products [87,88].

*2.4. Climate Change Demands Forward-Thinking*

2.4.1. Forestry is Based on Site Conditions—and These Are Changing

The "function" of an object, here of a (forest) ecosystem, is the task it has to fulfill (see boxed section). The general "functioning" of a certain ecosystem, however, depends on the site conditions to which it is exposed. The many site conditions interacting at each single spot in the landscape ("the site") can be summarized by three terms: (i) geomorphology (e.g., elevation above sea level, exposition, inclination), (ii) soil (e.g., geological bedrock, pH, base saturation, water and nutrient storage capacity), and (iii) climate (temperature and precipitation regimes). During the 20th century, foresters implicitly saw "the site" as a constant, at least in the life span of a human or tree. This assumption is surely correct for geomorphology and geology, but it is not correct for climate. Since the 1970s, the global annual temperature has been increasing by 0.17 K per decade, with an overall increase of ~0.85 K. The average global temperature (land and sea surface) for 2016 was 0.94 K; higher than the 20th century mean [89]. In the study region of Bavaria, the temperature increased by 1.4 K from 1881 to 2014 [90], and our own unpublished data for central Bavaria show an increase of 0.8 K within the last three decades. Thus, temperature is increasing strongly. Climate is not a constant and site conditions are changing within the life span of a human and of a tree.

Excellent knowledge on the site conditions is essential for a successful forest management. What is the relevance of changing climate for forestry? The concept of "potential natural vegetation" (PNV), outlined by Tüxen [91], uses the plant communities (usually forest communities) to express the set of site conditions, which is realized at a certain spot; the name of the communities is the symbol for this special site type. The Federal State of Bavaria (Southeast Germany), at about one third of the size of Shaanxi Province, was used as a test region to model a map of the PNV, separated into about 28 million pixels of 50 m to 50 m each. For each pixel, the most probable site-type was used [92]. In this model, it is possible to change site conditions, for example, by increasing temperature. The results are seismic: In the "+1K"-scenario (without change in precipitation), while all site types of the basis scenario remain, many spots experience site-type changes. Of note, "+1K" is not a scenario of the future but is what we

already have today (see above). In the "+2K"-scenario, on one third of the area of Bavaria, site types will occur that are not yet known here, and for the "+3K"-scenario this is true for two thirds of Bavaria. These results forecast that the fundamental basis of forest management, the best knowledge of site conditions, is going to disappear. Organizing multifunctional forest management in a sustainable way is complicated in itself, and climate change is thus not making it any easier. Incorporating climate change into forest planning will therefore be of particular importance.

### 2.4.2. Effects of Restoration Practices on Controlling Soil and Water Losses—A Case Study in the Wei River Catchment

With intensification of land use as urban expansion and climate change, surface vegetation destruction, and soil degradation and soil erosion have been exacerbated, leading to significant losses in agricultural production and weakening the capacity of soil to stored and recycled carbon, nutrients, and water, which has been seen as a serious problem for the ecological environment of the Wei River Catchment, Shaanxi Province, China [93,94].

The Wei River was notorious for its high erosion rates. Since the middle of the 20th century, however, the runoff (RO) and sediment discharge (SD) of the Wei River have decreased tremendously. This shift is well reflected in the measurements taken at the Hua County Observatory, a gateway field station where the Wei River outflows into the Yellow River. The annual RO and SD were $70.65 \times 10^8$ m$^3$ and 355 million tons, respectively, from 1970–1990. In comparison, these indicators declined to $50.04 \times 10^8$ m$^3$ and 113 million tons per year during 2000–2015. In other words, the average RO decreased by about 29.2%, while the SD dropped by 68.3% (Bureau of Hydrology of the Yellow River Conservancy Commission, personal communication 2015). Therefore, an important set of questions is: What are the drivers that have caused the reductions in RO and SD in the Wei River Catchment? Have they resulted simply from climate change and other natural factors, such as precipitation [95] or have they come from human interventions, such as implementing the Sloping Land Conversion Program (SLCP) [96,97]? If they have been affected by both natural and socioeconomic factors, to what extent have these factors contributed to the reductions? To address these questions, it is necessary and relevant to elucidate the relationships between RO, sediment concentration (SC), and SD and to capture the potential effects of those primary natural and socioeconomic determinants. Building on an interdisciplinary approach, this research will enable an integrative, systematic, and objective assessment of the effects of treatment measures, leading to more effective policy making and, ultimately, ensuring a more balanced, sustainable development in the catchment [98–102].

After adopting a simultaneous equations model, based on panel data covering 44 counties over a period of 16 years (2000–2015), results show that structural linkages between soil and water runoff are properly quantified [100]. Additionally, SC reduction is largely driven by the SLCP for improving land cover. Terracing has played a positive role in mitigating RO and SD. It can be expected that the effect of the SLCP on ecosystem services may be even greater in the future as the ecological environment is further improved. In the SLCP, erosion reduction is the main task or function, but being implemented, timber production can increasingly become a second main task of this program.

### 2.5. Afforestation and Participation of the Local Population are Beneficial and Healthy

### 2.5.1. Rehabilitation of Degraded Land Ecosystems by Afforestation with Native Tree Species—A Case Study in the Qinling Mountains

The economic centers in the dry northern parts of China are suffering from water scarcity. To improve the situation, China launched the "South-to-North Water Diversion Project", which was targeted to bring water from the southern parts of China to the North. The southern region of Shaanxi Province was part of this project. Unfortunately, the water in this region was often unusable for human consumption, both for the local rural population as well as for urban people in Northern China, due to suspended solids and toxins. The undesirable ingredients in the water came from intensive agricultural land use in the mountainous regions, which caused degradation, widespread erosion, and flooding

problems. To overcome such problems, China started the largest afforestation program in the world. Especially in mountainous regions, such as in the Qinling Mountains, every endeavor has been made to afforest degraded land. A joint project on rehabilitation of degraded land ecosystems by afforestation with native tree species by Chinese and German forest scientists had the objective to develop sustainable silvicultural methods to establish stable close-to-nature forest ecosystems [103]. The first results of an afforestation experiment with four native species (*Pinus tabulaeformis*, *Quercus variabilis*, *Acer truncatum*, *Pistacia chinensis*) at three sites in Shangnan County/Shaanxi Province indicate that afforestation can be an alternative to afforestation with exotic tree species [103]. The natural succession of abandoned farmland is a slow process, which is dominated by shrubs (mainly *Ziziphus jujuba*) in the early phase. Only a few trees, such as *Platycladus orientalis*, were present on the research sites. Without afforestation, it will take a long time for restoration of these sites, and the emerging new forests will probably not comply with the requirements of sustainable forestry.

After two years of observation, the survival and growth of the planted saplings of the four native tree species differed among the three sites. In spite of these variations, the results of this study demonstrate that native tree species can be used for the reforestation of abandoned farmland successfully. With the exception of *Quercus variabilis*, all species demonstrated a significantly better diameter growth by removing the ground vegetation, while no effect of the treatment on the height growth of each species could be observed. Considering costs and effects of weed control, a frequent removal of the ground vegetation can be neglected for these species. The repeated application of the biological fertilizer, RhizoVital® 42 fl., including a plant growth-promoting rhizobacterium, which is mainly applied in agriculture and horticulture, showed no significant effect, neither on the survival rate nor on the height or diameter growth of the four tree species. Therefore, fertilization with RhizoVital® 42 fl. of tree saplings cannot be recommended due to its high material and labor costs.

### 2.5.2. The Natural Forest Protection Program in Shaanxi Province

The Protection Program of Natural Forest Resources (PPNFR) was officially implemented in Shaanxi province in 2000, aiming to prevent the environmental degradation caused by excessive deforestation before 1998. Different approaches were adopted in the program to protect local forest resources, including prohibiting the commercial exploration of natural forests, closing hillsides for livestock grazing, comprehensively designating areas for nature reserves and national parks, and continuously promoting the intelligent process of natural forest supervision and monitoring.

During the past 20 years, the PPNFR was applied to all forested land in the province. The total protected area has reached 14.3 million ha, accounting for about 70% of the total land area in the province. About 2.4 million ha of public welfare forest and 0.5 million ha of young and middle-aged forests are cultivated. Approximately 5.2 billion Yuan (RMB) have been invested for ecological benefits compensation. In addition, the program also expanded into the development of other fields, such as forest tourism, underwood planting, e-commerce, forest health care, and eco-civilization education.

Implementing the PPNFR has achieved outstanding results. Most significantly, the forest resources in Shaanxi province, particularly the natural forests, have obtained an overall and continuous protection, resulting in a stable and extensive restoration of forest ecosystems. Additionally, the stock of living trees in forests doubled. In detail, the forest stock growth increased from originally 8.8 million $m^3$/year between 2000 and 2010 to 16.2 million $m^3$/year between 2011 and 2014. Furthermore, the protection of forests conserves about 42.6 billion $m^3$/year surface runoff water for 4300 rivers with a minimum watershed of 10 $km^2$. In this case, the main function of the forest ecosystem is to produce high quality drinking water, and indeed, until September 2018, more than 10 billion $m^3$ of water had been supplied.

### 2.5.3. Poverty Reduction and Forestry—A Case Study in the South of Shaanxi Province, China

Poverty reduction is defined as a collective responsibility to fight all avoidable forms of deprivation, make poor people less poor, enable poor people to escape from poverty, and build institutions and societies that prevent people from becoming poor or from slipping further into poverty [104]. Over

the past 40 years in China, 700 million poor people were able to escape poverty, which is over 70% of global poor people that have been lifted out of poverty in the same period [105]. However, by the end of 2017, there are still more than 30 million rural people living in poverty in China, and that will be the objective for poverty reduction by the end of 2020 [106].

Forests have always played an important role in poverty reduction through supporting income-generating activities, underpinning subsistence economies, providing a safety net to minimize vulnerability to risk, supplying poor rural household energy requirements, maintaining productivity of poor farmers' land use systems through applying different patterns of agroforestry practice, and integrated capacity building [107]. A case study concerning poverty reduction practice through forestry was conducted in Shaanxi, China [108]. The results showed that particularly the following approaches were helpful for poor people to get out of poverty in China: (i) By active participation in forest ecological projects to obtain ecological service compensation due to maintaining plantations or plantation establishment and by obtaining government subsidies due to acting as forest guards; (ii) by harvesting non-timber forest products for income generation, such as by operating tea plantations or cultivating chestnuts and walnuts or collecting cork (see also Section 2.6.1); and (iii) by development of a non-timber forest-based economy, such as by bee-keeping, cultivating herbal medicine and mushrooms, and operating businesses associated with local forest-based eco-tourism.

Forests thus directly or indirectly contribute to poverty reduction in terms of income generation, awareness-raising, and behavioral change. The inclusion of the poor in the generation of forest multifunctionality thereby forcefully elaborates the remark by President Xi Jinping: "Green mountains with clean water ARE mountains of gold and silver".

### 2.6. Forest Products—Ecologic and Economic Uses Can Overlap

### 2.6.1. Cork as a Multifunctional Forestry Resource Made in China

Cork is a secondary protection measure of oak trees and anatomically a part of the periderm. It is a natural and closed-cell bio-material with a set of excellent properties, such as low specific gravity and low permeability to gases and liquids, admirable thermal and sound insulation, wear-resistance, durability, and chemical stability [109,110].

In China, cork is mainly obtained from *Quercus variabilis* Blume, which is a wide-spread oak tree. The horizontal distribution of *Q. variabilis* in China is mainly in areas with 22° N and 42° N latitude and 99° E and 122° E longitude. There are about 22 provinces in China which have *Q. variabilis* cork resources. In China, the annual output of cork is about 50,000 tons. The cork yield of Shaanxi province accounts for about 50% of China [111,112].

Since the quality of *Q. variabilis* cork is lower than *Q. suber* [113], it cannot be used for the production of solid cork, such as for wine stoppers. Instead, it is granulated and used in agglomerates for various applications, such as cork sheets, note boards, shoe material, thermal insulation material, and decorative material.

In recent years, there have been a number of studies on *Q. variabilis* cork, which focused on the fields of structural characteristics, chemical components, physical properties, and basic applications. In the future, research on resource cultivation and high value-added applications of *Q. variabilis* cork should be enhanced in China, as well as technology innovation of the cork industry. A standardized system of a *Q. variabilis* cork industry should be set up in China as soon as possible. As a natural and renewable resource with special structure and chemical composition, cork can therefore be part of a multifunctional forestry, with a protective role in oak tree functioning and particular interest as a source of chemicals and bioproducts.

### 2.6.2. Wood—A Classical Forestry Product as Multifunctional Driver of the Modern Forest-Based Bioeconomy

The concept of a bioeconomy strives for a transition from current economic models based on the exploitation of fossil resources to more sustainable bio-based models [114]. Forests represent the world's largest production system of lignocellulosic biomass that is not in conflict with human nutrition. Among the many services offered by forests, the provision of round and fuel wood is the most meaningful and obligates the forest-based industries to take a leading role in the development of bioeconomy [115]. However, to achieve sustainability, resource availability and raw material demand need to be aligned, which is a challenge for both silvicultural management and forest products industries in the advent of a growing population and climate change. The population growth combined with an increased awareness of climate issues will give rise to a strongly growing demand of wood-based products, for example in the building sector, since wood can replace many traditional, carbon-intensive building materials, such as concrete and steel [116]. Climate change, however, will greatly impact local species composition (see also Section 2.4.1). For Germany, a higher proportion of hardwoods is anticipated [117], which will require the development of novel utilization and processing schemes. Sustainability furthermore demands an increase in the efficient use of all wood-based products. Therefore, products should be fit for their purpose, durable, reliable, and aesthetically attractive. In addition, the overall lifecycle of the wooden material itself needs to be extended over the product lifetime, which can be done by cascading use practices. When, for instance, "design-to-reuse" principles are integrated, products can be dismantled more easily out of their prime constructions and the recovered wood can be reused in further product cycles. Examples are solid laminated wood products (e.g., floorings), particle-based products (e.g., wood composites), fiber-based products (e.g., insulation board, paper, and cardboard), (bio-)chemical products (sugars, lignin, nano-cellulose, etc.), and finally energy products (e.g., heat) [118]. Our recent material flow modeling studies have elucidated that wood cascading has positive effects on selected environmental parameters [119] and impacts positively on resource efficiency [120] and eco-efficiency [121].

Biorefinery applications at the stage of bio-chemical wood utilization are also a topic of active investigation. Fungi, as natural decomposers of plant biomass or their enzymes, are used more and more for the liberation of sugars from recalcitrant fibers [122]. However, the efficiency of deconstruction varies depending on the composition, including lignin content and cellulose crystallinity as key determinants [123,124]. Moreover, cellulase production in several fungi is inhibited by mannan, being the major hemicellulose in softwoods [125]. An ideal microorganism would not only utilize multiple available carbon sources in wood but would also have a low sensitivity to inhibitors and would be capable of producing many metabolic products. More research is therefore needed to align the capabilities of microorganisms used for biorefinery to the necessities of the varying woody feedstocks.

### 2.6.3. Combinatorial Wood Modifications Can Help to Extend its Service Life

The Chinese Natural Forest Protection project was initiated in 1998. In recent years, the main native timber resources in China are fast-growing plantations. The main plantation species are poplar (*Populus* L.) in North China, eucalyptus (*Eucalyptus robusta* Smith), and masson pine (*Pinus massoniana* Lamb.) in South China.

There are some shortcomings in the application of the fast-growing plantation timbers, such as low strength, low hardness, poor decay resistance, and poor dimensional stability. Wood is a porous viscoelastic material, which can be compressed to improve the hardness and strength at high temperature and high pressure without destroying the structure of the cell wall.

The combination of glycerol pretreatment [126,127], boron pretreatment, and wood thermal treatment can improve the dimensional stability of wood and accelerate the thermal degradation of wood cell wall components [128,129]. Moreover, the combination of boron pretreatment and heat treatment can improve the decay resistance and mold resistance of wood [130,131]. Combinatory

modifier and thermal treatment methods can thus help to substantially prolong the service life of wood and thereby contribute to the function of wood as a carbon storage material.

*2.7. Multifunctionality on a Supra-Regional Scale*

Management of the Qinling Mountains

For the Chinese nation, nature was very generous to give two mother rivers—the Yellow River and the Yangtze River, as well as the Qinling Mountains as the father mountain. This is one of the most exquisite geographical combinations on Earth, constructing the most unique "one mountain, two rivers" zone in the world.

China highly values ecological and environmental protection. Guided by the conviction that lucid waters and lush mountains are invaluable assets of nature, the country advocates coexistence between humans and nature and sticks to the path of green and sustainable development [132]. The Qinling Mountains represent the "lucid waters and lush mountains" at the core of China.

People are familiar with the "four treasures of the Qinling Mountains": Giant pandas, crested ibis, golden monkeys, and takins. In a broader sense, the four treasures of the Qinling Mountains are water, forest, culture, and beauty; all hallmarks of multifunctionality. The Qinling Mountain is the green reservoir in the heart of China. Qinling water supports more than one-tenth of the country's population. The Qinling forest is a treasure island of Chinese forests, a gene pool of species, and one of the key areas of biodiversity protection [133]. From Zhou to Tang dynasties, Chang'an and Luoyang, the former capital cities of the Chinese Empire, stood at the foot of the Qinling Mountains for two thousand years. The Qinling Mountains are among the most beautiful sceneries in China. The natural beauty and the cultural beauty complement each other and became the name card of "Chinese style".

From the perspective of the whole Chinese nation, the Qinling Mountains are thus seen as the core of China. In fact, Qinling is located in Qinghai, Gansu, Sichuan, Shaanxi, Chongqing, Hubei, and Henan provinces, and for the six provinces and one city, the Qinling Mountains are on the edge of each province. The region is poor and underdeveloped. To use Qinling's forests in a multifunctional manner, it is therefore necessary to establish a special institution for protecting, repairing, and shaping the Qinling ecological environment. The example also demonstrates that aspects like beauty, tradition, and national identity are part of multifunctionality and have to be included in an overall planning scheme for multifunctional land use.

## 3. Conclusions

*3.1. Multifunctionality of Forests as a Target for a Modern, Sustainable Forestry*

This paper outlines the general concept of multifunctionality in forests and, based on case studies, throws light on the many aspects involved in multifunctionality. Wrapping up, we want to highlight several issues of importance for implementing multifunctionality as a guiding idea of further forest management. These issues have been worked out for forest landscapes in the temperate zone of Central Europe and Eastern China, but they are formulated in a wide sense, so that they can be generalized for forests worldwide.

*3.2. Multifunctionality: A General Concept, Which has to be Locally Adapted*

Forest area is limited and demands are many: Wood is quality timber, fire wood, a basis for paper, pulp, and chemicals, water storage, soil erosion prevention, carbon sequestration, air cleaning, landscape structuring, a place for recreation, tradition, and national identity, but also a job offer, and therefore a survival guarantee in many regions, especially for poor people. Thus, several or many functions are assigned to each spot in the landscape. All of these functions have to be included in landscape managing planning processes. Social agreements have to be reached for setting priorities, including economic, ecological, and mental issues and valuation. The list of issues and their priority

will change from place to place, but everywhere the whole system has to be kept in mind, not only focusing on one (or a few) issues.

### 3.3. Biodiversity is the Basis

Biodiversity is not seen here as a function of the forest ecosystem, but it is the fundamental basis of forest ecosystem functioning and stability. Therefore, it has to be carefully handled and protected everywhere in the course of forest management.

### 3.4. Forest Management for the Future: Sustainability in the Focus

Most of the resources on Earth are limited, and this is also true for forest resources. Utilisation of these resources in a way that will also give future generations the same possibilities is therefore mandatory. Both the concept of "sustainability", developed in Western culture, and the concept of an "eco-civilization", developed in China, aim at this goal.

### 3.5. Knowledge-Based Forest Planning

Forest management following the multifunctionality concept requires multi-criteria analyses and scenarios. On the one hand, options for future utilisation of forest resources have to be scientifically analysed. On the other hand, the consequences of either continued forest exploitation (business as usual) or introduction of new management practices (new target materials, new management processes, new species) have to be carefully studied especially regarding feedbacks to the other functions. Long term monitoring plots help to understand steady-state forest growth, economic analyses demonstrate financial challenges and options, sociological studies make demands of society beyond economy visible. Altogether, we need well-educated forest scientists and forest practitioners, not only today but also tomorrow: We need high-standard forest education at both universities and universities of applied sciences.

### 3.6. Trees are Slow-Growing—Foresight and Long Wind are Needed

While new agricultural decisions on which species to plant and how to care for them are taken each year, trees grow for decades or even centuries. Therefore, decisions made in forest management are binding for at least half a century. As a consequence, the planning procedure is much more complicated than in agriculture and needs a lot of knowledge far beyond how to plant and harvest a tree, a synoptic view, enough time, and, most crucially, enough education.

### 3.7. Forestry between Ecology, Economy, and Cultural Expectancy

Very often, ecological and economic goals, as well as cultural expectations, are in conflict. It is the task of silviculture to develop utilization concepts that reconcile the different demands, minimize the conflicts, and optimize the overall societal contentment regarding forest land use. Within the concept, single forest operations also need to be pro-active and planned beforehand according to the aspired functions or products, respectively. As long as they are in line with the natural site conditions, mixed forests seem to be better than single-species forests in the sense of resilience and resistance against calamities and disasters.

### 3.8. Climate Change Demands Forward Thinking

While the environmental background of land use has been seen as mostly constant up till now, we know that this is no longer true. Due to climate change, trees planted today will grow and finally be harvested in a different (warmer/dryer) environment. Thus, in forest management, not only the current site conditions, biodiversity, demands of society have to be included in a farsighted land use planning system, but also the environmental changes to be expected between planting and harvesting trees. Similarly, of course, the demands of society may change. Anticipating such trends on a scientific

basis and including them in long-term forest management planning is among the most important services we can do to following generations.

### 3.9. Multiple Actors and Multiple Demands on a Limited Resource—A Friction Zone

To find a way through the jungle of demands, system restrictions (e.g., ecological benchmarks), economic interests, and societal expectations, a well working system of networking, communication, and transparency needs to be established and fostered. To enable the next generation's stakeholders to face the complex challenges, they need excellent education in system-thinking. Economic evaluations may form one basis for negotiations among interest groups, but economy is not the only thing that people need; e.g., "beauty of nature", "health", or "national identity" can hardly be expressed in terms of money and cannot be compared to profit from timber selling.

### 3.10. Implementation of Multifunctionality is a Permanent Process

Resources are limited, demands are increasing, and environmental and societal framework conditions are changing. As a consequence, implementation of the idea of multifunctionality is a rather more permanent process than reaching a distinct final stage. In Germany, this path has been followed for several decades. China is now on the way to it. Optimization of the whole forest or even a landscape system in terms of contributing to a multitude of human demands is the overarching goal instead of maximizing one single demand. This also has to include future developments, such as climate or social change; therefore, sustainability and multifunctionality are two sides of one single coin.

**Author Contributions:** All authors contributed to the conception of the work and to the acquisition of data. A.F., J.P.B. and H.P. drafted the original manuscript. All authors contributed text to the manuscript and revised it critically for important intellectual content. All authors have read and agreed to the published version of the manuscript.

**Funding:** Funding for the workshop was provided jointly by a TUM Global Incentive Fund project grant, the Northwest A&F University, and the FEDRC GIZ Forest Policy Facility. The funding bodies had no role in the design of the workshop nor in the collection, analysis, and interpretation of data discussed or in writing of the manuscript.

**Acknowledgments:** We would like to thank the representatives of Northwest A&F University for perfectly organizing the workshop in Yangling. We also would like to thank Zhenshan Jin (TUM Liaison Office Beijing) for help in the organization of the workshop. The following people are gratefully acknowledged for the critical reading of the text and assistance in the described studies: Peter Biber, Barbara Michler, Hagen S. Fischer, Michael Suda, Shan Sun, Liyan Liang, Ge Liang, Luxi Jiang, Bernhard Felbermeier, Jörg Summa, Hany El Kateb, Li Jia, and Zhai Xiao Jiang.

**Conflicts of Interest:** The authors declare that they have no competing interests.

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
