# Peer review of "Multifunctionality of Forests: A White Paper on Challenges and Opportunities in China and Germany"

_forests, doi:10.3390/f11030266_

Round 1
Reviewer 1 Report
The manuscript was generally well-updated based on the comments. The abstract is informative, the conference name was shown in the text, some citations were added, and the text was partly reshaped for reducing the similarity to the recent cited literature. This paper is informative and fruitful for those who interested in the multifunctionality of forests.
Author Response
Answer: We thank this reviewer for his/her input and believe that by the additions and modifications based on his/her feedback the manuscript has become much stronger.
Reviewer 2 Report
Rows 88-90. Storing CO2 does not fight the causes, which are mainly due to very different reasons such as fuels' emissions, industry, traffic, livestock farms, and so on ...
Rows 105-106. Multifunctionality is a spontaneous feature of forest ecosystems. Rather, it needs only to be better known, valued, and computed as an ordinary and varied tool in environmental economics. In parallel, forest restoration is frequently a necessary tool to enhance or recover acceptable level of multifunctionality.
Par. 2.1.1 A more exhaustive comparison between high forest systems and the coppicing systems is necessary. Moreover, protection for flooding and soil erosion, mitigation of temperature extremes (and means), water storing, release, and supply, CO2 sink functions, oxygen production, and biodiversity conservation need to be implemented as indispensable environmental and economic functions today.
In general, an interesting article. Although it is coherent through its development, it looks more appropriate as a chapter of a good book in these fields than a scientific paper. It covers a wide range of relevant issues, however the scientific analysis tends to be insufficient to cover all the topics introduced especially in terms of adaption forest planning and management capacity.
Author Response
Rows 88-90. Storing CO2 does not fight the causes, which are mainly due to very different reasons such as fuels' emissions, industry, traffic, livestock farms, and so on ...
Answer: We agree. The sentence was modified accordingly (“the causes” was deleted).
Rows 105-106. Multifunctionality is a spontaneous feature of forest ecosystems. Rather, it needs only to be better known, valued, and computed as an ordinary and varied tool in environmental economics. In parallel, forest restoration is frequently a necessary tool to enhance or recover acceptable level of multifunctionality.
Answer: The paragraph was modified to include the point of multifunctionality being per se “spontaneous” (or inherent) and the fact that correct valuation needs to be performed and communicated.
Part. 2.1.1. A more exhaustive comparison between high forest systems and the coppicing systems is necessary. Moreover, protection for flooding and soil erosion, mitigation of temperature extremes (and means), water storing, release, and supply, CO2 sink functions, oxygen production, and biodiversity conservation need to be implemented as indispensable environmental and economic functions today.
Answer: This part develops a new theoretical perspective on the issues mentioned by the reviewer. We now highlight the increasing demands on forests and the new developments (short-rotation plantations on agricultural land, increased demand for agroforestry systems) at the end of Part. 2.1.1.
In general, an interesting article. Although it is coherent through its development, it looks more appropriate as a chapter of a good book in these fields than a scientific paper. It covers a wide range of relevant issues, however the scientific analysis tends to be insufficient to cover all the topics introduced especially in terms of adaption forest planning and management capacity.
Answer: We are grateful for the positive remark. Indeed, the topic of multifunctionality demands a thorough scientific discussion, since the network of all functions and actors is complex, but highly relevant. However, and as already eluded to in our cover letter, the current manuscript, as a white paper, was rather meant to provide an overview regarding the multitude of functions and their interactions for a broad audience. This unfortunately comes at the expense of depth. Indeed, there would be enough material available for a book, but this is a good idea for a future effort. For further information, we are referring the interested reader to the cited literature. Moreover, for this reason, we think that the manuscript is a perfect addition to the current special issue topic: “Decision Support to Address Multiple Ecosystem Services in Forest Management Planning”, since the reader can find in-depth reports on many touched-on topics directly flanking the article.
Round 2
Reviewer 2 Report
With regard to rows 207-213, reporting 2-3 examples of the total balance between carbon emissions/release and carbon storage in coppices in the medium-long term would be very desirable.
Author Response
Dear reviewer,
thanks for the final remark. We have incorporated this suggestion with a small paragraph on the issue exemplifying with two new citations the carbon-related potential (and challenge) of coppicing (lines 206-213 in the track-changes version).
We hope that the manuscript is now acceptable in its current form. Thanks again for all the constructive feedback!
This manuscript is a resubmission of an earlier submission. The following is a list of the peer review reports and author responses from that submission.
Round 1
Reviewer 1 Report
General comments:
This paper summarized a joint workshop on multi-functionality of forests among researchers in China and Germany in 2018, showing various aspects relating the topics, which is informative to the readers who interested in forest multifunctionality. I found a few points for improving the quality of the manuscript.
(1) The abstract can be improved by a concise summarization of various topics described in this paper. Whereas, it is better to remove topics that were not shown in depth in the manuscript, such as “smart forestry”.
(2) It is better to write in short the official name of the workshop and approximated number of participants etc. e.g., after L 112. It would be informative for the readers.
(3) I found copy-and-paste of previous literature or websites, in some sections particularly from 2.2.2 to 2.5.1, of which a few of them have no citations. In order not to be plagiarism, the authors should rewrite them by summarizing concisely. Those parts are as follows:
L329-340: similar to the abstract of Lin et al 2017 without citation (http://www.publish.csiro.au/bt/Fulltext/bt16233?subscribe=false)
L344-353: there is some similarity with Yuan et al 2017a.
L354-368: there is some similarity with Yuan et al 2017b.
L376-380, L384-395, L398-406: there is some similarity with Sun et al 2019.
L421-426, L430-434, L437-450: Many similar sentences with Pretzsch et al 2019.
L454-457, L468-473: some similarities with Khanna et al 2018
L525-549: many similar sentences with Wang and Yao 2019.
L555-563: This is a copy of the following website without citation, http://china.wzw.tum.de/index.php?L=1&id=69
Minor comments
L 248 and 266: B -> billion. (It is better to be consistent e.g., similar to the sections 2.5.2 and 2.5.3.)
L 293: kg m2 -> kg m-2
L 306: It is better to spell out “m3/PMH0” in the text.
L 384: represent?
L 465, 466: system approaches, system approach -> systems approach
L 566: remove space between sentences.
L 672: aesthetically
Reviewer 2 Report
This is a nice perspective on multifunctionality of forest as the title indicated. The report tries to make recommendations to policy makers about how to manage forests locally or regionally. But, the concept can be applied anywhere in the world. It is a well-written manuscript. I enjoy reading it.
Some comments for authors to consider:
L43: Throughout the manuscript, citations are not formatted to Forests. I wonder if this is acceptable since editor sent out for review.
L58: Is there reason for excluding building materials from the list?
L66: Wildlife habitats are one of major ecosystem services, but was not mentioned in the manuscript. It is ironic that authors proudly mentioned four treasures of the Qinling Mountains: giant pandas, crested ibis, golden monkeys and takinsthere. It seems me that protecting their habitats should have been one of major management goals.
L87: “Fulfil” should have been “fulfill”.
L94: Rapid rate of climate change.
L96: Change to: … forest planners to …
L147: Again, the wildlife habitats was excluded.
L158: … without compromising or comprising?
L234-238: Where are these numbers from?
L241: This statement cannot be true. Perhaps there is no other functions stated in the planning document. Or some functions are not studied or demonstrated in the paper. Lack of statements in the document does not mean that there is non-existing. For example, commercial forests will also prevent the soil erosion, will not?
L329: Who are we?
L382: Too wordy here – suggest deleting “as a matter of fact”.
L487: Fulfil should have been fulfill.
L494: Should sine be since?
L499: Climate has never been a constant for the past million years. It is the change rate that makes different right now.
L545: Watch for the sentence structure. Who show …
L546: Delete “based on simultaneous equations model”.
L557: A period, not semicolon, after “situation”.
L573: Without seed source on the abandoned farmland, where are the secondary forests from? This is so obvious for a forester or any common-sense people. Please rephrase it!
L576: This could be a laughable study if the objective is to test whether native species can be established by planting them. Perhaps just state that native species have been well established on these farmlands and growth …
L580: What elements are in this biological fertilizer? What rate had been applied? How old was the plantation? I can’t believe the firm conclusion stated holds true for any situation on these lands.
L588: Are you referring animals to wildlife or livestock?
L608: Should “potable water” be “portable water”?
L618: What is an intimate tree-crop or tree-livestock interactions?
L717: The sentence is a duplication of the previous sentence.
L737: How can we manage regional forests to construct an environment?
L764 and 804: Suggest changing “the following generations” to “future generations”.